# GaussianFlow: Splatting Gaussian Dynamics for 4D Content Creation

**Quankai Gao**                                                    *quankaig@usc.edu*
*University of Southern California*

**Qiangeng Xu**                                                    *qiangenx@google.com*
*Google*

**Zhe Cao**                                                        *Zhecao@google.com*
*Google*

**Ben Mildenhall**                                                 *bmild@google.com*
*Google*

**Wenchao Ma**                                                     *wmm5390@psu.edu*
*Pennsylvania State University*

**Le Chen**                                                        *le.chen@tuebingen.mpg.de*
*Max Planck Institute for Intelligent Systems*

**Danhang Tang**                                                   *danhangtang@google.com*
*Google*

**Ulrich Neumann**                                                 *uneumann@usc.edu*
*University of Southern California*

**Reviewed on OpenReview:** *https://openreview.net/forum?id=XBL7xi5rt0*

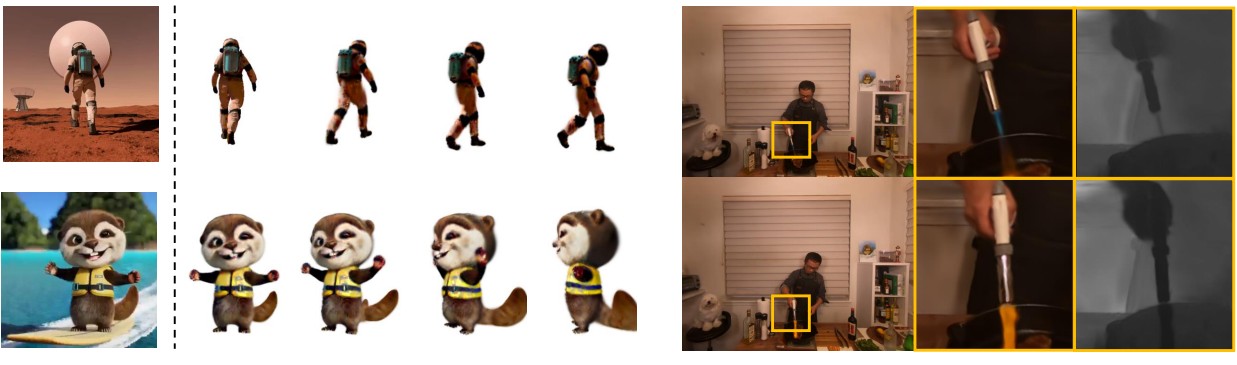

(a) 4D Generation                                    (b) 4D Novel View Synthesis

Figure 1: We propose GaussianFlow, a dense 2D motion flow created by splatting 3D Gaussian dynamics, which significantly benefits tasks such as 4D generation and 4D novel view synthesis. (a) Based on monocular videos generated by Lumiere Bar-Tal et al. (2024) and Sora Brooks et al. (2024), our model can generate 4D Gaussian Splatting fields that represent high-quality appearance, geometry and motions. (b) For 4D novel view synthesis, the motions in our generated 4D Gaussian fields are smooth and natural, even in highly dynamic regions where other existing methods suffer from undesirable artifacts.

## Abstract

Creating 4D fields of Gaussian Splatting from images or videos is a challenging task due to its under-constrained nature. While the optimization can draw photometric reference from the input videos or be regulated by generative models, directly supervising Gaussian motions remains underexplored. In this paper, we introduce a novel concept, Gaussian flow, which connects the dynamics of 3D Gaussians and pixel velocities between consecutive frames. The Gaussian flow can be obtained efficiently by splatting Gaussian dynamics into the image space. This differentiable process enables direct dynamic supervision from optical flow. Our method significantly benefits 4D dynamic content generation and 4D novel view synthesis with Gaussian Splatting, especially for contents with rich motions that are hard to handle by existing methods. The common color drifting issue that occurs in 4D generation is also resolved with improved Guassian dynamics. Superior visual quality in extensive experiments demonstrates the effectiveness of our method. As shown in our evaluation, GaussianFlow can drastically improve both quantitative and qualitative results for 4D generation and 4D novel view synthesis. Project page: https://zerg-overmind.github.io/GaussianFlow.github.io/

## 1 Introduction

4D dynamic content creation from monocular or multi-view videos has garnered significant attention from academia and industry due to its wide applicability in virtual reality/augmented reality, digital games, and movie industry. Studies (Li et al., 2022; Pumarola et al., 2021; Park et al., 2021a;b) model 4D scenes using 4D dynamic Neural Radiance Fields (NeRFs) and optimize them based on input multi-view or monocular videos. Once optimized, the 4D field can be viewed from novel camera poses at preferred time steps through volumetric rendering. A more challenging task is generating 360 degree 4D content based on uncalibrated monocular videos or synthetic videos generated by text-to-video or image-to-video models. Since monocular input cannot provide enough multi-view cues and unobserved regions are not supervised due to occlusions, studies (Singer et al., 2023; Jiang et al., 2023; Zhao et al., 2023) optimize 4D dynamic NeRFs by leveraging generative models to create plausible and temporally consistent 3D structures and appearance. The optimization of 4D NeRFs requires volumetric rendering, which makes the process time-consuming. And real-time rendering of optimized 4D NeRFs is also hardly achieved without special designs. A more efficient alternative is to model 4D radiance fields by 4D Gaussian Splatting (GS) (Wu et al., 2023; Luiten et al., 2023), which extends 3D Gaussian Splatting (Kerbl et al., 2023) with a temporal dimension. Using the efficient rendering of 3D GS, the lengthy training time of a 4D Radiance Field can be drastically reduced (Yang et al., 2023c; Ren et al., 2023) and rendering can achieve real-time speed during inference.

The optimization of 4D Gaussian fields takes photometric loss as the major supervision. As a result, the scene dynamics are usually under-constraint. Similarly to 4D NeRFs (Li et al., 2023; Park et al., 2021a; Pumarola et al., 2021), the radiance properties and the time-varying spatial properties (location, scales, and orientations) of Gaussians are both optimized to reduce the photometric Mean Squared Error (MSE) between the rendered frames and the input video frames. The ambiguities of appearance, geometry, and dynamics have been introduced in the process and become prominent with sparse-view or monocular video input. Per-frame Score Distillation Sampling (SDS) (Tang et al., 2023) reduces the appearance-geometry ambiguity to some extent by involving multi-view supervision in the latent domain. However, both monocular photometric supervision and SDS supervision do not directly supervise scene dynamics.

To avoid temporal inconsistency caused by fast motions, Consistent4D (Jiang et al., 2023) leverages a video interpolation block, which imposes a photometric consistency between the interpolated frame and the generated frame, at the cost of involving more frames as pseudo ground truth for fitting. Similarly, AYG (Ling et al., 2023) uses the text-to-video diffusion model to balance motion magnitude and temporal consistency with a preset frame rate. 4D NeRF model (Li et al., 2023) has proven that optical flows in reference videos are strong motion cues and can significantly benefit scene dynamics. However, for 4D GS, connecting 4D Gaussian motions with optical flows has the following two challenges. First, a Gaussian motion is in 3D space, but it is its 2D splat that contributes to rendered pixels. Second, multiple 3D Gaussians might contribute to the same pixel in rendering, and each pixel's flow does not equal to any one Gaussian's motion.

To overcome these challenges, we introduce a novel concept, Gaussian flow, bridging the dynamics of 3D Gaussians and pixel velocities between consecutive frames. Specifically, we assume that the optical flow of each pixel in the image space is influenced by the Gaussians that cover it. The Gaussian flow of each pixel is considered to be the weighted sum of these Gaussian motions in 2D. To obtain the Gaussian flow value on each pixel without losing the speed advantage of Gaussian Splatting, we splat 3D Gaussian dynamics, including scaling, rotation, and translation in 3D space, onto the image plane along with its radiance properties. As the whole process is end-to-end differentiable, the 3D Gaussian dynamics can be directly supervised by matching Gaussian flow with optical flow on input video frames. We apply such flow supervision to both 4D content generation and 4D novel view synthesis to showcase the benefit of our proposed method, especially for contents with rich motions that are hard to handle by existing methods. The flow-guided Guassian dynamics also resolve the color drifting artifacts that are commonly observed in 4D Generation. We summarize our contributions as follows:

- We introduce a novel concept, Gaussian flow, that first bridges the 3D Gaussian dynamics to resulting pixel velocities, enabling flow supervision for Gaussian Splatting based representations. Matching Gaussian flows with optical flows, 3D Gaussian dynamics can be directly supervised.

- The Gaussian flow can be obtained by splatting Gaussian dynamics into the image space. Following the tile-based design by original 3D Gaussian Splatting, we implement the dynamics splatting in CUDA with minimal overhead. The operation to generate dense Gaussian flow from 3D Gaussian dynamics is highly efficient and end-to-end differentiable.

- With Gaussian flow to optical flow matching, our model drastically improves over existing Gaussian Splatting based-methods, especially on scene sequences of fast motions. Color drifting is also resolved with our improved Gaussian dynamics.

## 2 Related Works

**3D Generation.** 3D generation has attracted tremendous attention with the progress of various 2D or 3D-aware diffusion models (Liu et al., 2023b; Rombach et al., 2022; Shi et al., 2023b; Liu et al., 2023c) and large vision models Radford et al. (2021); Jun & Nichol (2023); Nichol et al. (2022). Thanks to the availability of large-scale multi-view image datasets (Deitke et al., 2023; Yu et al., 2023; Downs et al., 2022), object-level multi-view cues can be encoded in generative models and are used for generation purpose. Pioneered by DreamFusion (Poole et al., 2022) that first proposes Score Distillation Sampling (SDS) loss to lift realistic content from 2D to 3D via NeRFs, 3D content creation from text or image input has flourished. This progress includes approaches based on online optimization (Tang et al., 2023; Lin et al., 2023; Wang et al., 2024c; Raj et al., 2023) and feedforward methods (Hong et al., 2023; Liu et al., 2023a; 2024; Xu et al., 2023; Wang et al., 2023c) with different representations such as NeRFs Mildenhall et al. (2021), triplane (Chan et al., 2022; Chen et al., 2022; Gao et al., 2023) and 3D Gaussian Splatting (Kerbl et al., 2023). 3D generation becomes more multi-view consistent by involving multi-view constraints (Shi et al., 2023b) and 3D-aware diffusion models (Liu et al., 2023b) as SDS supervision. Not limited to high-quality rendering, studies (Sun et al., 2023; Long et al., 2023) also explore enhancing the quality of generated 3D geometry by incorporating normal cues.

**4D Novel View Synthesis and Reconstruction.** By adding timestamp as an additional variable, recent 4D methods with different dynamic representations such as dynamic NeRF (Park et al., 2021a;b; Li et al., 2021; Wang et al., 2023a; Li et al., 2022; Tretschk et al., 2021; Gao et al., 2021), dynamic triplane Fridovich-Keil et al. (2023); Cao & Johnson (2023); Shao et al. (2023) and 4D Gaussian Splatting Huang et al. (2024); Wu et al. (2023); Yang et al. (2023c); Lin et al. (2024); Yang et al. (2024); Duan et al. (2024); Guo et al. (2024) are proposed to achieve high quality 4D motions and scene contents reconstruction from either calibrated multi-view or uncalibrated RGB monocular video inputs. Some works (Newcombe et al., 2011; 2015; Zollhöfer et al., 2014) reconstruct rigid and non-rigid scene contents with RGB-D sensors, which help to resolve 3D ambiguities by involving depth cues. Different from static 3D reconstruction and novel view synthesis, 4D novel view synthesis consisting of both rigid and non-rigid deformations is notoriously challenging and ill-posed with only RGB monocular inputs. Some progress (Li et al., 2021; Gao et al.,

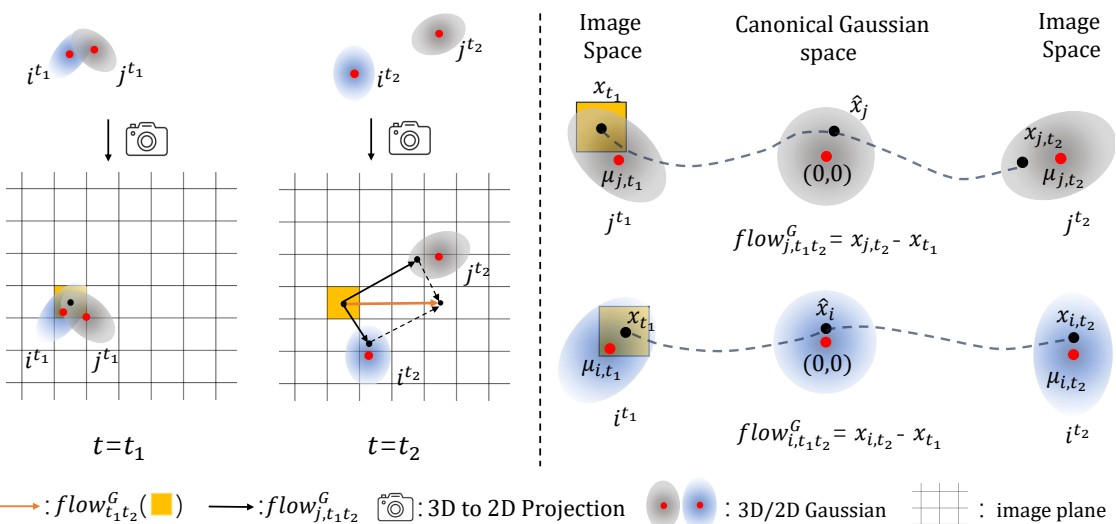

Figure 2: We illustrate our GaussianFlow formulation using two Gaussian primitives. **Flow Composition (Left)**: the two Gaussians, indexed by $i$ and $j$ are shown at two consecutive time steps $t_1$ and $t_2$. The Gaussian flow on the yellow pixel $x_{t_1}$ is defined as the weighted sum of the shift contributions from all Gaussians ($i$ and $j$) covering the pixel. The weighting factor utilizes alpha composition weights the same as in the volume rendering. The Gaussian flow of the entire image can be obtained efficiently by splatting 3D Gaussian dynamics and rendering with alpha composition. **Flow from Single Gaussian (Right)**: Between two consecutive frames, the yellow pixel $x_{t_1}$ will be pushed towards $x_{t_1} \to x_{i,t_2}$ by the 2D Gaussian $i$'s motion $i^{t_1} \to i^{t_2}$, where we denote this shift contribution from Gaussian $i$ as $flow_{i,t_1,t_2}^G$. We can track $x_{t_1}$ in Gaussian $i$ by normalizing it to canonical Gaussian space as $\hat{x}_i$ and unnormalize it to image space to obtain $x_{i,t_2}$. A similar process applies for Gaussian $j$.

2021; Tretschk et al., 2021; Wang et al., 2021; 2025) involves temporal priors and motion cues (e.g. optical flow) to better regularize temporal photometric consistency and 4D motions. One of recent works (Wang et al., 2023a) provides an analytical solution for flow supervision on deformable NeRF without inverting the backward deformation function from the world coordinate to the canonical coordinate. Several works (Yang et al., 2021a;b; 2023a;b) explore object-level mesh recovery from monocular videos with optical flow.

**4D Generation.** Similar to 3D generation from text prompts or single images, 4D generation from text prompts or monocular videos also relies on frame-by-frame multi-view cues from pre-trained diffusion models. Besides, 4D generation methods yet always rely on either video diffusion models or video interpolation block to ensure the temporal consistency. Animate124 (Zhao et al., 2023), 4D-fy (Bahmani et al., 2023) and one of the earliest works Singer et al. (2023) use dynamic NeRFs as 4D representations and achieve temporal consistency with text-to-video diffusion models, which can generate videos with controlled frame rates. Instead of using dynamic NeRF, Align Your Gaussians (Ling et al., 2023) DreamGaussian4D (Ren et al., 2023) and L4GM Ren et al. (2024) generate vivid 4D contents with 3D Gaussian Splatting, but again relying on video diffusion models for free frame rate control. Without the use of text-to-video diffusion models, Consistent4D (Jiang et al., 2023) achieves coherent 4D generation with an off-the-shelf video interpolation model (Huang et al., 2022). Our method benefits 4D Gaussian representations by involving flow supervision and without the need of specialized temporal consistency networks.

## 3 Methodology

To better illustrate the relationship between Gaussian motions and corresponding pixel flow in image space, we first recap the rendering process of 3D Gaussian Splatting and then investigate its 4D case.

### 3.1 Preliminary

**3D Gaussian Splatting.** From a set of initialized 3D Gaussian primitives, 3D Gaussian Splatting aims to recover the 3D scene by minimizing the photometric loss between input $m$ images $\{I\}_m$ and rendered images $\{I_r\}_m$. For each pixel, its rendered color $C$ is the weighted sum of multiple Gaussians' colors $c_i$ in depth order along the ray by point-based $\alpha$-blending as in Eq. 1,

$$C = \sum_{i=1}^{N} T_i \alpha_i c_i, \tag{1}$$

with weights specifying as

$$\alpha_i = o_i e^{-\frac{1}{2}(\mathbf{x}-\boldsymbol{\mu}_i)^T \boldsymbol{\Sigma}_i^{-1}(\mathbf{x}-\boldsymbol{\mu}_i)} \quad \text{and} \quad \prod_{j=1}^{i-1}(1 - \alpha_j). \tag{2}$$

where $o_i \in [0, 1]$, $\boldsymbol{\mu}_i \in \mathbb{R}^{2 \times 1}$, and $\boldsymbol{\Sigma}_i \in \mathbb{R}^{2 \times 2}$ are the opacity, 2D mean, and 2D covariance matrix of $i$-th 2D Gaussian projected from 3D space, respectively. And $\mathbf{x}$ is the intersection between a pixel ray and $i$-th Gaussian. As shown in Eq. 1, the relationship between a rendered pixel and 3D Gaussians is not bijective.

**3D Gaussian Splatting in 4D.** Modeling 4D motions with 3D Gaussian Splatting can be done frame-by-frame via either directly multi-view fitting (Luiten et al., 2023) or moving 3D Gaussians with a time-variant deformation field (Ling et al., 2023; Ren et al., 2023) or parameterizing 3D Gaussians with time (Yang et al., 2023c). While with monocular inputs, Gaussian motions are under-constrained because different Gaussian motions can lead to the same rendered color, and thus long-term persistent tracks are lost (Luiten et al., 2023). Though Local Rigidity Loss (Luiten et al., 2023; Ling et al., 2023) is proposed to reduce global freedom of Gaussian motions, it sometimes brings severe problems due to poor or challenging initialization and lack of multi-view supervision. As shown in Fig. 6, 3D Gaussians initialized with the skull mouth closed are hard to be split when the mouth open with Local Rigidity Loss.

### 3.2 GaussianFlow

We consider the full freedom of each Gaussian motion in a 4D field, including 1) scaling, 2) rotation, and 3) translation at each time step. As time changes, Gaussians covering the queried pixel at $t = t_1$ will move to other places at $t = t_2$, as shown in Fig. 2(left). To specify new pixel location $\mathbf{x}_{t_2}$ at $t = t_2$, we first project all the 3D Gaussians into 2D image plane as 2D Gaussians and calculate their motion's influence on pixel shifts.

**Flow from Single Gaussian.** To track pixel shifts (flow) contributed by Gaussian motions, we let the relative position of a pixel in a deforming 2D Gaussian remain the same. This setting preserves the mahalanobis distance between the pixel locations under two consecutive time steps and the 2D Gaussian unchanged. According to Eq. 2, this preservation will grant the pixel with the same radiance and $\alpha$ contribution from the 2D Gaussian, albeit the 2D Gaussian is deformed.

The pixel shift (flow) is the image space distance of the same pixel at two time steps. We first calculate the pixel shift influenced by a single 2D Gaussian that covers the pixel. As shown in Fig. 2(right), we can find a pixel $\mathbf{x}$'s location at $t_2$ by normalizing its image location at $t_1$ to canonical Gaussian space and unnormalizing it to the image space in $t_2$:

1) *normalize.* A pixel $\mathbf{x}_{t_1}$ following $i$-th 2D Gaussian distribution can be written as $\mathbf{x}_{t_1} \sim N(\boldsymbol{\mu}_{i,t_1} \boldsymbol{\Sigma}_{i,t_1})$. And in $i$-th Gaussian coordinate system with 2D mean $\boldsymbol{\mu}_{i,t_1} \in \mathbb{R}^{2 \times 1}$ and 2D covariance matrix $\boldsymbol{\Sigma}_{i,t_1} \in \mathbb{R}^{2 \times 2}$. After normalizing the $i$-th Gaussian into the standard normal distribution, we denote the pixel location in canonical Gaussian space as

$$\hat{\mathbf{x}}_{t_1} = \mathbf{B}_{i,t_1}^{-1}(\mathbf{x}_{t_1} - \boldsymbol{\mu}_{i,t_1}), \tag{3}$$

which follows $\boldsymbol{\Sigma}_{i,t_1} = \mathbf{B}_{i,t_1}\mathbf{B}_{i,t_1}^T$, $\hat{\mathbf{x}}_{t_1} \sim N(\mathbf{0}, \mathbf{I})$ and $\mathbf{I} \in \mathbb{R}^{2 \times 2}$ is identity matrix.

2) *unnormalize*. When $t = t_2$, the new location along with the Gaussian motion denotes $\mathbf{x}_{i,t_2}$ on the image plane.

$$\mathbf{x}_{i,t_2} = \mathbf{B}_{i,t_2}\hat{\mathbf{x}}_{t_1} + \boldsymbol{\mu}_{i,t_2}, \tag{4}$$

and $\boldsymbol{\Sigma}_{i,t_2} = \mathbf{B}_{i,t_2}\mathbf{B}_{i,t_2}^T$, $\mathbf{x}_{t_2} \sim N(\boldsymbol{\mu}_{i,t_2}, \boldsymbol{\Sigma}_{i,t_2})$. Eq. 3 and Eq. 4 preserve the Mahalanobis distance between the tracked pixel and the 2D Gaussian, leading to a consistent value of $\alpha$ (see Eq.2) for this pixel over consecutive time steps. The pixel shift contribution from each Gaussian therefore can be calculated as:

$$flow_{i,t_1t_2}^G = \mathbf{x}_{i,t_2} - \mathbf{x}_{t_1} \tag{5}$$

**Flow Composition.** In the original 3D Gaussian Splatting, a pixel's color is the weighted sum of the 2D Gaussians' radiance contribution. Similarly, we define the Gaussian flow value at a pixel as the weighted sum of the 2D Gaussians' contributions to its pixel shift, following alpha composition. With Eq. 3 and Eq. 4, the Gaussian flow value at pixel $\mathbf{x}_{t_1}$ from $t = t_{t_1}$ to $t = t_{t_2}$ is

$$flow_{t_1t_2}^G = \sum_{i=1}^{K} w_i flow_{i,t_1t_2}^G \tag{6}$$

$$= \sum_{i=1}^{K} w_i \left[ \mathbf{B}_{i,t_2}\mathbf{B}_{i,t_1}^{-1}(\mathbf{x}_{t_1} - \boldsymbol{\mu}_{i,t_1}) + \boldsymbol{\mu}_{i,t_2} - \mathbf{x}_{t_1} ) \right], \tag{7}$$

where $K$ is the number of Gaussians along each camera ray sorted in depth order and each Gaussian has weight $w_i = \frac{T_i\alpha_i}{\Sigma_i T_i\alpha_i}$ according to Eq. 1, but normalized to [0,1] along each pixel ray. The intuition behind the use of the same weight as $\alpha$-blending is that, if a pixel color is contributed by a weighted sum of a set of Gaussians, then its corresponding pixel shift i.e. pixel-wised optical flow should also be contributed by the same set of Gaussians with the same weights by nature, since optical flow is calculated based on the pixel-wised correspondences as well.

In some cases Ling et al. (2023); Keetha et al. (2023); Yugay et al. (2023); Matsuki et al. (2023), each Gaussian is assumed to be isotropic, and its scaling matrix $\mathbf{S} = \sigma\mathbf{I}$, where $\sigma$ is the scaling factor. And its 3D covariance matrix $\mathbf{RSS}^T\mathbf{R}^T = \sigma^2\mathbf{I}$. If the scaling factor of each Gaussian does not change too much over time, $\mathbf{B}_{i,t_2}\mathbf{B}_{i,t_1}^{-1} \approx \mathbf{I}$. Therefore, to pair with this line of work, the formulation of our Gaussian flow as in Eq. 7 can be simplified as

$$flow_{t_1t_2}^G = \sum_{i=1}^{K} w_i(\boldsymbol{\mu}_{i,t_2} - \boldsymbol{\mu}_{i,t_1}). \tag{8}$$

In other words, for isotropic Gaussian fields, Gaussian flow between two different time steps can be approximated as the weighted sum of individual translation of 2D Gaussian.

Following either Eq. 7 or Eq. 8, the Gaussian flow can be densely calculated at each pixel. The flow supervision at pixel $\mathbf{x}_{t_1}$ from $t = t_1$ to $t = t_2$ can then be specified as

$$\mathcal{L}_{flow} = ||flow_{t_1t_2}^o(\mathbf{x}_{t_1}) - flow_{t_1t_2}^G||, \tag{9}$$

where optical flow $flow_{t_1t_2}^o$ can be calculated by off-the-shelf methods as pseudo ground-truth. Our method also allows for camera motions, please refer to our experiments on NeRF-DS dataset (Yan et al., 2023) for more details.

### 3.3 4D Content Generation

As shown in Fig. 3, 4D content generation with Gaussian representation takes an uncalibrated monocular video either by real capturing or generating from text-to-video or image-to-video models as input and output a 4D Gaussian field. 3D Gaussians are initialized from the first video frame with photometric supervision between rendered image and input image and a 3D-aware diffusion model (Liu et al., 2023b) for multi-view

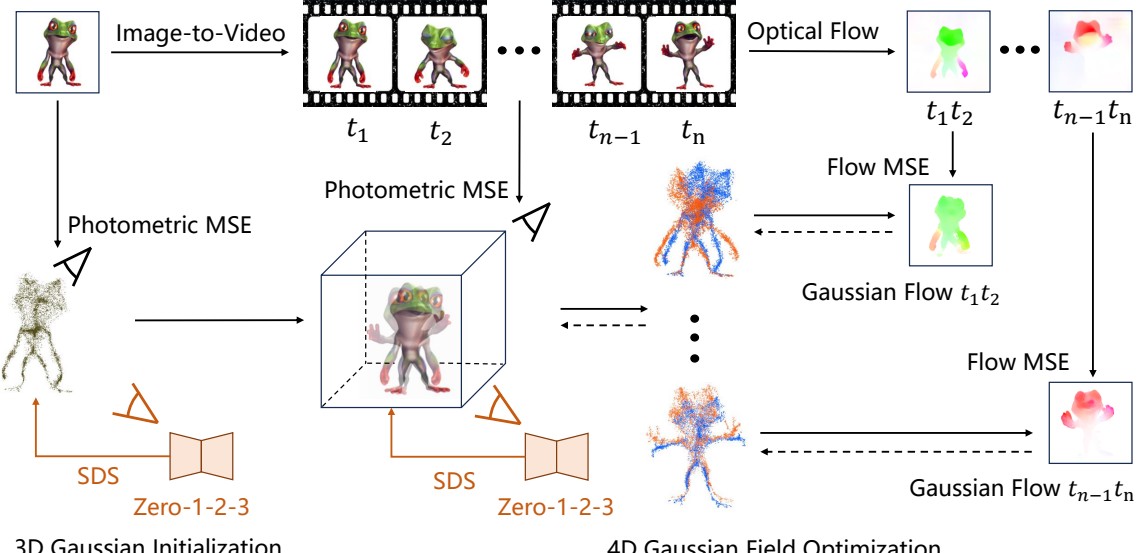

Figure 3: Overview of our 4D content generation pipeline. An uncalibrated monocular video or video generated from an image is taken as the input. We optimize a 3D Gaussian field initialized by the first frame with both photometric and SDS supervision (Liu et al., 2023b) (for 4D generation only). Then, we optimize the dynamics of the 3D Gaussians with the same two losses for each frame. Most importantly, we calculate Gaussian flows with our novel design on reference view for each consecutive two time steps and match it with a pre-computed optical flow of the input video. The gradients from the flow matching will propagate back through dynamics splatting and rendering process, resulting in a 4D Gaussian field with natural and smooth motions.

**SDS supervision.** In our method, 3D Gaussian initialization can be done by One-2-3-45 (Liu et al., 2024) or DreamGaussian (Tang et al., 2023). After initialization, a 4D Gaussian field is optimized with per-frame photometric supervision, per-frame SDS supervision, and our flow supervision as in Eq. 9. The loss function for 4D Gaussian field optimization can be written as:

$$\mathcal{L} = \mathcal{L}_{photometric} + \lambda_1 \mathcal{L}_{flow} + \lambda_2 \mathcal{L}_{sds} + \lambda_3 \mathcal{L}_{other}, \tag{10}$$

where $\lambda_1$, $\lambda_2$ and $\lambda_3$ are hyperparameters. $\mathcal{L}_{other}$ is optional and method-dependent. Though not used in our method, we leave it for completeness.

### 3.4 4D novel view Synthesis

Unlike 4D content generation that has multi-view object-level prior from the 3D-aware diffusion model, 4D novel view synthesis takes only multi-view or monocular input video frames for photometric supervision without any scene-level prior. 3D Gaussians are usually initialized by sfm (Snavely et al., 2006; Schonberger & Frahm, 2016) from input videos. After initialization, the 4D Gaussian field is optimized with per-frame photometric supervision and our flow supervision. We adopt the 4D Gaussian field from (Yang et al., 2023c). The loss function for 4D Gaussian field optimization can be written as:

$$\mathcal{L} = \mathcal{L}_{photometric} + \lambda_1 \mathcal{L}_{flow} + \lambda_2 \mathcal{L}_{other}, \tag{11}$$

where $\mathcal{L}_{other}$ is optional and method-dependent (please refer to Yang et al. (2023c)).

## 4 Experiments

In this section, we first provide implementation details of the proposed method and then validate our method on 4D Gaussian representations with (1) 4D novel view synthesis and (2) 4D generation. We tested on

the Plenoptic Video Datasets (Li et al., 2022) and the Consistent4D Dataset (Jiang et al., 2023) for both quantitative and qualitative evaluation. Our method achieves state-of-the-art results in both tasks. To obtain dense Gaussian flow, we efficiently splatting the Gaussian dynamics along with the original 3DGS(Kerbl et al., 2023) CUDA pipeline. Please refer to our supplemental materials for implementation details.

## 4.1 Dataset

**Plenoptic Video Dataset.** A high-quality real-world dataset consists of 6 scenes with 30FPS and 2028 × 2704 resolution. There are 15 to 20 camera views per scene for training and 1 camera view for testing. The cameras are distributed to face the frontal part of scenes from different angles.

**NeRF-DS Dataset.** This dataset (Yan et al., 2023) consists of 8 scenes in everyday environments with various types of moving or deforming specular objects. Each scene contains two videos captured by two rigidly mounted forward-facing cameras.

**Consistent4D Dataset.** This dataset (Jiang et al., 2023) includes 14 synthetic and 12 in-the-wild monocular videos. All the videos have only one moving object with a white background. 7 of the synthetic videos are provided with multi-view ground-truth for quantitative evaluation. Each input monocular video with a static camera is set at an azimuth angle of 0°. Ground-truth images include four distinct views at azimuth angles of -75°, 15°, 105°, and 195°, respectively, while keeping elevation, radius, and other camera parameters the same with input camera.

## 4.2 Results and Analysis

**4D Novel View Synthesis.** We visualize rendered images and depth maps of a very recent state-of-the-art 4D Gaussian method RT-4DGS (Yang et al., 2023c) with (yellow) and without (red) our flow supervision in Fig. 4. According to zoom-in comparisons, our method can consistently model realistic motions and correct structures. These regions are known to be challenging (Verbin et al., 2022; Liu et al., 2023d) for most methods, even under adequate multi-view supervision. Our method can reduce ambiguities in photometric supervision by involving motion cues and is shown to be consistently effective across frames. By using an off-the-shelf optical flow algorithm (Shi et al., 2023a), we found that only a small portion of image pixels from the Plenoptic Video Dataset have optical flow values larger than one pixel. Since our method benefits 4D Gaussian-based methods more on the regions with large motions, we report PSNR numbers on both full scene reconstruction and dynamic regions (optical flow value > 1) in Tab. 1. With the proposed flow supervision, our method shows improved performance on all scenes and the gains are prominent on dynamic regions. Consequently, our 4D novel view synthesis results achieve state-of-the art quality. Both qualitative and quantitative comparisons on the NeRF-DS dataset in Fig. 10 and Tab. 2 show the effectiveness of the proposed method on scenes with complex camera motions. We also report PSNR, D-SSIM, LPIPS averaged over all scenes of the DyNeRF dataset (Li et al., 2022) in Tab. 5. More comparisons are shown in the Fig. 11-13 and the video in our supplemental material.

**4D Generation.** We evaluate and compare our method with DreamGaussian4D (Ren et al., 2023), a recent 4D Gaussian-based state-of-the-art generative model with open-sourced code, and dynamic NeRF-based methods D-NeRF (Pumarola et al., 2021), K-planes (Fridovich-Keil et al., 2023) and Consistent4D (Jiang et al., 2023) in Tab. 3 on Consistent4D dataset. Scores on individual videos are calculated and averaged over four novel views mentioned above. Note that flow supervision is effective and helps with 4D generative Gaussian representation. Compared to DreamGaussian4D, our method shows better quality as shown in Fig. 6 after the same number of training iterations. For the two challenging cases shown in Fig. 6, our method benefits from flow supervision and generates desirable motions, while DG4D shows prominent artifacts on the novel views. Furthermore, flow supervision helps our method avoid color drifting, compared with the dynamic NeRF-based method Consistent4D(Jiang et al., 2023) (Fig. 5). Our results are more consistent in terms of texture and geometry. We also show more generation results in Fig. 8 of the supplemental material.

Table 1: Quantitative evaluation between ours and other methods on the DyNeRF dataset Li et al. (2022). We report PSNR numbers on both full-scene novel view synthesis and dynamic regions where the ground-truth optical flow value is larger than one pixel. "Ours" denotes RT-4DGS with the proposed flow supervision. We also achieve the best results on D-SSIM and LPIPS (see the Tab. 5 and 4 in the supplemental material).

| Method | Coffee Martini | Spinach | Cut Beef | Flame Salmon | Flame Steak | Sear Steak | Mean |
|---|---|---|---|---|---|---|---|
| HexPlane Cao & Johnson (2023) | - | 32.04 | 32.55 | 29.47 | 32.08 | 32.39 | 31.70 |
| K-Planes Fridovich-Keil et al. (2023) | **29.99** | 32.60 | 31.82 | 30.44 | 32.38 | 32.52 | 31.63 |
| MixVoxels Wang et al. (2023b) | 29.36 | 31.61 | 31.30 | 29.92 | 31.21 | 31.43 | 30.80 |
| NeRFPlayer Song et al. (2023) | 31.53 | 30.56 | 29.35 | **31.65** | 31.93 | 29.12 | 30.69 |
| HyperReel Attal et al. (2023) | 28.37 | 32.30 | 32.92 | 28.26 | 32.20 | 32.57 | 31.10 |
| 4DGS Wu et al. (2023) | 27.34 | 32.46 | 32.90 | 29.20 | 32.51 | 32.49 | 31.15 |
| RT-4DGS Yang et al. (2023c) | 28.33 | 32.93 | 33.85 | 29.38 | 34.03 | 33.51 | 32.01 |
| Ours | 28.42 | **33.68** | **34.19** | 29.37 | **34.22** | **34.06** | **32.32** |
| Dynamic Region Only | | | | | | | |
| RT-4DGS Yang et al. (2023c) | 27.36 | 27.47 | 34.48 | 23.16 | 26.04 | 29.52 | 28.00 |
| Ours | **28.02** | **28.71** | **35.18** | **23.36** | **27.53** | **31.14** | **28.99** |

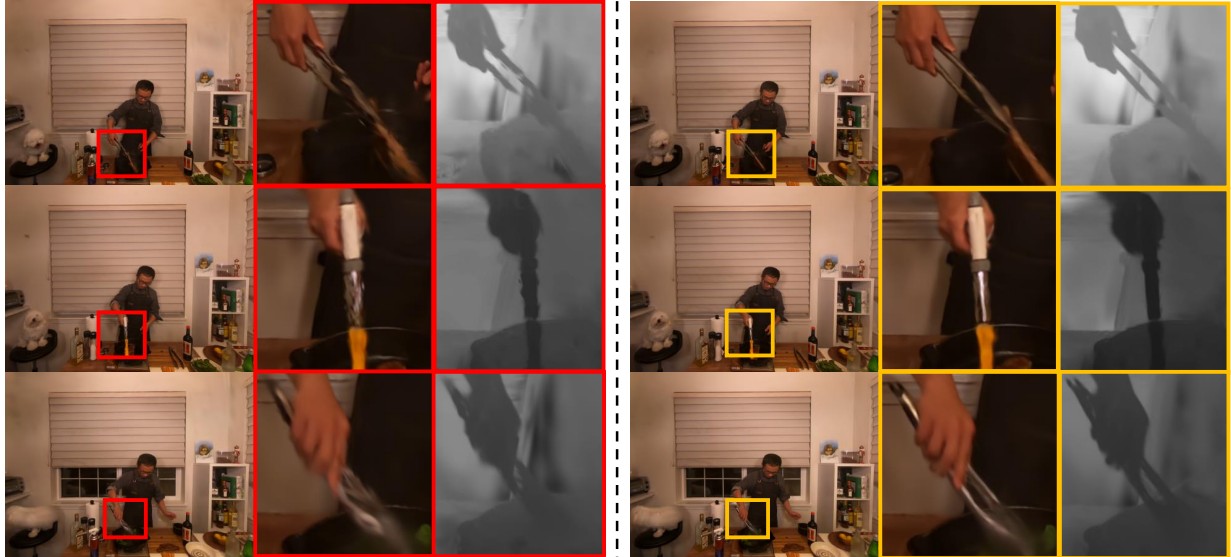

Figure 4: Qualitative comparisons on DyNeRF dataset (Li et al., 2022). The left column shows the novel view rendered images and depth maps of RT-4DGS (Yang et al., 2023c), which suffers from artifacts in the dynamic regions. The right column shows the results of RT-4DGS optimized with our flow supervision during training. We refer to our supplementary material (Fig. 11-13, including the video) for more visual comparisons.

## 5 Ablation Study

We validate our flow supervision through qualitative comparisons shown in Fig. 6. Compared with Ours (no flow) and Ours, the proposed flow supervision shows its effectiveness on moving parts. For the skull, 3D Gaussians in the teeth region initialized at $t = t_1$ are very close to each other and it is hard to split apart completely when $t = t_2$ as Gaussians can move freely as long as they look photometrically correct from view 0, while SDS supervision applied from novel views works on latent domains and cannot provide pixel-wised supervision. This problem becomes more severe when involving Local Rigidity Loss (comparing Ours-r and Ours) because the motions of 3D Gaussians initialized at $t = t_1$ are constrained by their neighbors and the Gaussians are harder to split apart at $t = t_1$. Similarly, for bird, regions consisting of thin structures such as the bird's beak cannot be perfectly maintained across frames without our flow supervision. While originally utilized in 4D Gaussian fields (Luiten et al., 2023) to maintain structure consistency during motion, Local

Table 2: Quantitative comparisons on NeRF-DS dataset (Yan et al., 2023). "Ours" denotes Deformable-3DGS with the proposed flow supervision. Please refer to section D. of our supplementary material for more details and qualitative results.

|  | PSNR ↑ | SSIM ↑ | LPIPS↓ |
|---|---|---|---|
| 3DGS (Kerbl et al., 2023) | 20.79 | 0.78 | 0.29 |
| TiNeuVo (Fang et al., 2022) | 21.60 | 0.83 | 0.28 |
| HyperNeRF (Park et al., 2021b) | 23.45 | **0.85** | 0.19 |
| NeRF-DS (Yan et al., 2023) | 23.60 | **0.85** | 0.18 |
| Deformable-3DGS (Yang et al., 2024) | 23.89 | 0.84 | 0.18 |
| Ours | **24.23** | **0.85** | **0.17** |

Table 3: Quantitative comparisons between ours and others on Consistent4D dataset.

| Method | Pistol | | Guppie | | Crocodile | | Monster | | Skull | | Trump | | Aurorus | | Mean | |
|---|---|---|---|---|---|---|---|---|---|---|---|---|---|---|---|---|
|  | LPIPS↓ | CLIP↑ | LPIPS↓ | CLIP↑ | LPIPS↓ | CLIP↑ | LPIPS↓ | CLIP↑ | LPIPS↓ | CLIP↑ | LPIPS↓ | CLIP↑ | LPIPS↓ | CLIP↑ | LPIPS↓ | CLIP↑ |
| D-NeRF | 0.52 | 0.66 | 0.32 | 0.76 | 0.54 | 0.61 | 0.52 | 0.79 | 0.53 | 0.72 | 0.55 | 0.60 | 0.56 | 0.66 | 0.51 | 0.68 |
| K-planes | 0.40 | 0.74 | 0.29 | 0.75 | 0.19 | 0.75 | 0.47 | 0.73 | 0.41 | 0.72 | 0.51 | 0.66 | 0.37 | 0.67 | 0.38 | 0.72 |
| Consistent4D | **0.10** | 0.90 | 0.12 | 0.90 | 0.12 | 0.82 | 0.18 | 0.90 | **0.17** | 0.88 | 0.23 | **0.85** | 0.17 | 0.85 | 0.16 | 0.87 |
| DG4D | 0.12 | 0.92 | 0.12 | 0.91 | 0.12 | 0.88 | 0.19 | 0.90 | 0.18 | 0.90 | 0.22 | 0.83 | 0.17 | 0.86 | 0.16 | 0.87 |
| Ours | **0.10** | **0.94** | **0.10** | **0.93** | **0.10** | **0.90** | **0.17** | **0.92** | **0.17** | **0.92** | **0.20** | **0.85** | **0.15** | **0.89** | **0.14** | **0.91** |

Table 4: SSIM, MSSIM, D-SSIM and LPIPS of our method on the DyNeRF dataset breakdown by scenes.

|  | Coffee Martini | Spinach | Cut Beef | Flame Salmon | Flame Steak | Sear Steak | Mean |
|---|---|---|---|---|---|---|---|
| SSIM ↑ | 0.9185 | 0.9578 | 0.9598 | 0.9248 | 0.9643 | 0.9645 | 0.9483 |
| MSSIM ↑ | 0.9544 | 0.9786 | 0.9808 | 0.9597 | 0.9816 | 0.9808 | 0.9726 |
| D-SSIM ↓ | 0.0228 | 0.0107 | 0.0096 | 0.0202 | 0.0092 | 0.0096 | 0.0137 |
| LPIPS ↓ | 0.0708 | 0.0389 | 0.0378 | 0.0639 | 0.0337 | 0.0354 | 0.0468 |

Table 5: Overall quantitative comparisions between ours and other methods on the DyNeRF dataset (Li et al., 2022). We report PSNR, D-SSIM, LPIPS averaged over all scenes. "Ours" denotes RT-4DGS with the proposed flow supervision.

|  | Mean PSNR ↑ | Mean D-SSIM ↓ | Mean LPIPS ↓ |
|---|---|---|---|
| HexPlane Cao & Johnson (2023) | 31.70 | **0.014** | 0.075 |
| K-Planes Fridovich-Keil et al. (2023) | 31.63 | 0.018 | - |
| MixVoxels Wang et al. (2023b) | 30.80 | 0.02 | 0.126 |
| NeRFPlayer Song et al. (2023) | 30.69 | 0.034 | 0.111 |
| HyperReel Attal et al. (2023) | 31.10 | 0.036 | 0.096 |
| 4DGS Wu et al. (2023) | 31.15 | 0.016 | 0.150 |
| RT-4DGS Yang et al. (2023c) | 32.01 | **0.014** | 0.055 |
| Ours | **32.32** | **0.014** | **0.047** |

Rigidity Loss as a motion constraint can incorrectly group Gaussians and is less effective than our flow supervision.

We also visualize optical flow $flow^o_{t_1 t_2}$ and Gaussian flow $flow^G_{t_1 t_2}$ with and without our flow supervision in Fig. 7. In both cases, the optical flow $flow^o_{t_1 t_2}$ between rendered images on the input view are very similar to each other (shown in #1 and # 4 column) and align with ground-truth motion because of direct photometric supervision on input view. However, comparing optical flows on novel view as shown in #3 and #6, without photometric supervision on novel views, inconsistent Gaussian motions are witnessed without our flow supervision. Gaussian flow $flow^G_{t_1 t_2}$ in #2 column also reveals the inconsistent Gaussian motions.

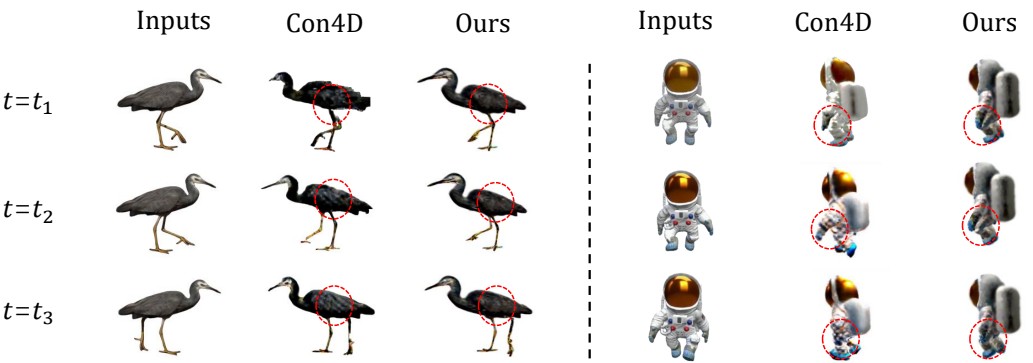

Figure 5: Comparisons between Consistent4D (Jiang et al., 2023) (a dynamic NeRF-based method) and ours. The flow supervision help us avoid the "bubble like" texture and non-consistent geometry on novel views.

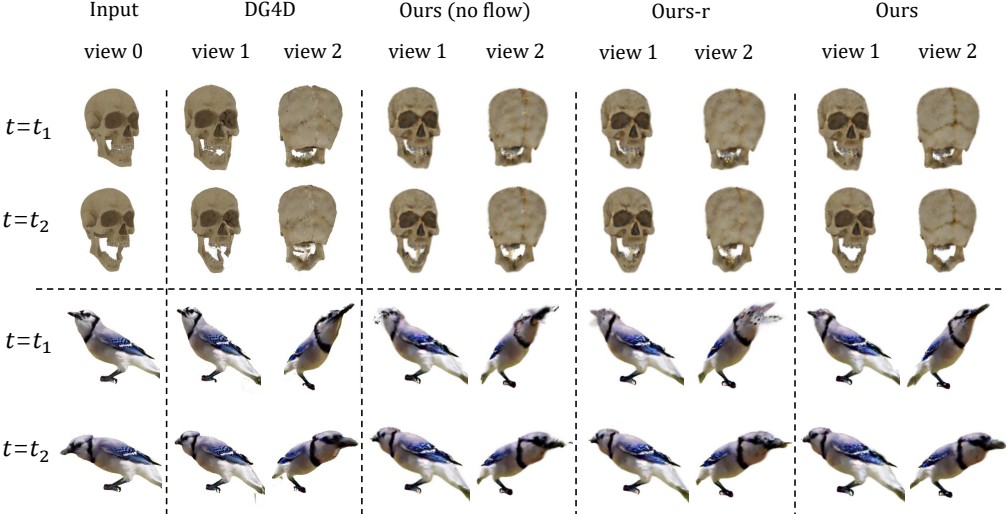

Figure 6: Qualitative comparisons among DreamGaussian4D (Ren et al., 2023), our method without flow loss, our method without flow loss but with Local Rigidity Loss (Ours-r) and ours.

Incorrect Gaussian motion can still hallucinate correct image frames on input view. However, this motion-appearance ambiguity can lead to unrealistic motions from novel views (the non-smooth flow color on moving parts in #3). While #5 shows consistent Gaussian flow, indicating consistent Gaussian motions with flow supervision.

# 6 More Discussions

By aligning with the optical flow, our Gaussian flow effectively optimizes the motion of Gaussian splats. However, if the optical flow cannot be reliably estimated, our method cannot provide a beneficial signal for optimization. For similar reason, this supervision is less helpful for modeling dynamic objects with constantly changing textures, which remains a challenge for current 4D generation methods. Unlike commonly used motion priors as optical flow, recent advances in dynamic scene reconstruction propose more compact and robust scene representations and regularizations, such as graph-based representations in MoSca (Lei et al., 2024), depth priors for 3D-aware tracking in SpatialTracker (Xiao et al., 2024), low-rank motion bases in Shape of Motion Wang et al. (2024a), and motion-aware point map representations in MonST3R (Zhang et al., 2024), leveraging Dust3R (Wang et al., 2024b). All of these techniques tightly couple the learnable

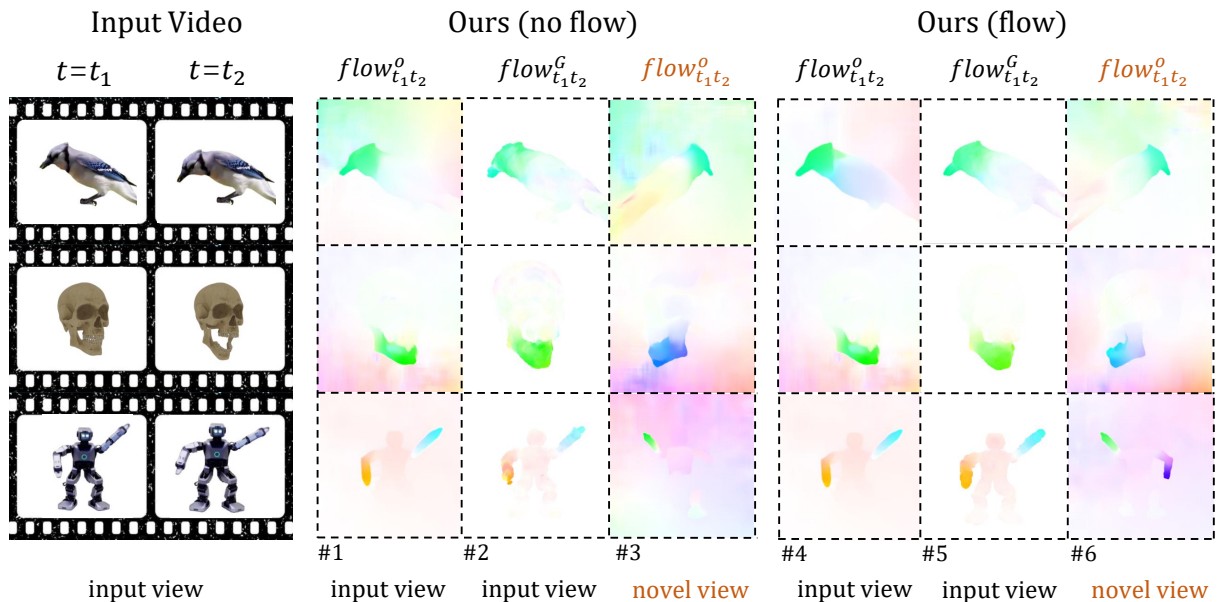

Figure 7: Visualization of optical and Gaussian flows on the input view and a novel view. "Ours (no flow)" denotes our model without flow supervision while "Ours" is our full model. The optical flow values of the background should be ignored because dense optical flow algorithms calculate correspondences among background pixels. We calculate optical flow $flow^o_{t_1 t_2}$ on rendered sequences by autoflow (Sun et al., 2021). From column #1 and #4, we can see that both rendered sequences from input view have high-quality optical flow, indicating correct motions and appearance. Comparing Gaussian flow $flow^G_{t_1, t_2}$ at column #2 and #5 , we can see that the underlining Gaussians move inconsistently without flow supervision. It is due to the ambiguity of appearance and motions while only being supervised by photometric loss on a single input view. Aligning Gaussian flow to optical flow can drastically improve irregular motions(column #3) and create high-quality dynamic motions (column #6) on novel views.

representations with the objective in an end-to-end fashion, making them more effective than integrating off-the-shelf optical flow into dynamic scene reconstruction, as done in our work. However, our proposed GaussianFlow can be seamlessly applied to any pixel-wise 2D-3D supervision, as our formulation establishes a general relationship between motions observed in 2D frames and the dynamics of 3D Gaussian primitives.

## 7 Conclusion and Future Work

We present GaussianFlow, an analytical solution for supervising 3D Gaussian dynamics including scaling, rotation, and translation with 2D optical flow. Extensive qualitative and quantitative comparisons demonstrate that our method is general and beneficial to Gaussian-based representations for both 4D generation and 4D novel view synthesis with motions. In this paper, we only consider the short-term flow supervision between every two neighbor frames in all our experiments. Long-term flow supervision across multiple frames is expected to be better and smoother, which we leave as future work. Another promising future direction is to explore view-conditioned flow SDS to supervise Gaussian flow on novel view in the 4D generation task.

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

# A  Appendix

## A.1  Implementation Details

We take $t_2$ as the next time step of $t_1$ and calculate the optical flow between every two neighbor frames in all experiments. In our CUDA implementation of Gaussian dynamics splatting, although the number of Gaussians $K$ along each pixel ray is usually different, we use $K = 20$ to balance speed and effectiveness. A larger $K$ means more number of Gaussians and their gradient will be counted through backpropagation. For video frames with size $H \times W \times 3$, we track the motions of Gaussians between every two neighbor timesteps $t_1$ and $t_2$ by maintaining two $H \times W \times K$ tensors to record the indices of the top-$K$ Gaussians sorted in depth order, top-$K$ Gaussians' rendered weights $w_i$ for each pixel and another tensor with size $H \times W \times K \times 2$ denotes the distances between the pixel coordinate and 2D Gaussian means $\mathbf{x}_{t_1} - \boldsymbol{\mu}_{i,t_1}$, respectively. In addition, the 2D mean $\boldsymbol{\mu}_{i,t_1}$ and 2D covariance matrices $\boldsymbol{\Sigma}_{i,t_1}$ and $\boldsymbol{\Sigma}_{i,t_2}$ of each Gaussian at different two timesteps are accessible via camera projection (Kerbl et al., 2023).

---

**Algorithm 1:** Detailed pseudo code for GaussianFlow

---

**Input:**

$flow^o_{t_k,t_{k+1}}$ : Pseudo ground-truth optical flow from off-the-shelf optical flow algorithm;

$I^{gt}_{t_k}$: ground-truth images , where $k = 0, 1, ..., T$;

*renderer*: A Gaussian renderer;

$Gaussians_{t_k}, Gaussians_{t_{k+1}}$ : $n$ Gaussians with learnable parameters at $t_k$ and $t_{k+1}$;

$cam_{t_k}$ and $cam_{t_{k+1}}$: Camera parameters at $t_k$ and $t_{k+1}$;

# Loss init

$\mathcal{L} = 0$

**for** timestep $k \leq T - 1$ **do**

    // renderer outputs at $t_k$

    $renderer_{t_k} = renderer(Gaussians_{t_k}, cam_{t_k})$;

    $I^{render}_{t_k} = renderer_{t_k}[\text{``image''}]$;    # $H \times W \times 3$

    $idx_{t_k} = renderer_{t_k}[\text{``index''}]$;    # $H \times W \times K$, Gaussian indices that cover each pixels

    $w_{t_k} = renderer_{t_k}[\text{``weights''}]$;    # $H \times W \times K$

    $w_{t_k} = w_{t_k}/sum(w_{t_k}, dim = -1)$;    # $H \times W \times K$, weight normalization

    $x\_\mu_{t_k} = renderer_{t_k}[\text{``x\_mu''}]$; # $H \times W \times K \times 2, denotes$  $x_{t_k} - \mu_{t_k}$

    $\mu_{t_k} = renderer_{t_k}[\text{``2D\_mean''}]$; # $n \times 2$

    $\Sigma_{t_k} = renderer_{t_k}[\text{``2D\_cov''}]$;    # $n \times 2 \times 2$

    $B_{t_k} = \Sigma^{\frac{1}{2}}_{t_k}$;

    # renderer outputs at $t_{k+1}$

    $renderer_{t_{k+1}} = renderer(Gaussians_{t_{k+1}}, cam_{t_{k+1}})$;

    $\mu_{t_{k+1}} = renderer_{t_{k+1}}[\text{``2D\_mean''}]$; # $n \times 2$

    $\Sigma_{t_{k+1}} = renderer_{t_{k+1}}[\text{``2D\_cov''}]$;    # $n \times 2 \times 2$

    $B_{t_{k+1}} = \Sigma^{\frac{1}{2}}_{t_k}$;

    # Eq.8 while ignoring resize operations for simplicity

    $flow^G_{t_k,t_{k+1}} = w_{t_k} * \left(B_{t_{k+1}}[idx_{t_k}] * inv(B_{t_k})[idx_{t_k}] * x\_\mu_{t_k} + (\mu_{t_{k+1}}[idx_{t_k}] - \mu_{t_k}[idx_{t_k}] - x\_\mu_{t_k})\right)$

    # Eq.10

    $\mathcal{L}_{flow} = norm(flow^o_{t_k,t_{k+1}}, sum(flow^G_{t_k,t_{k+1}}, dim = 0))$

    # (1) Loss for 4D novel view synthesis

    $\mathcal{L} = \mathcal{L} + \mathcal{L}_{photometric}(I^{render}_{t_k}, I^{gt}_{t_k}) + \lambda_1 \mathcal{L}_{flow} + \lambda_3 \mathcal{L}_{other}$

    # (2) Loss for 4D generation

    $\mathcal{L} = \mathcal{L} + \mathcal{L}_{photometric}(I^{render}_{t_k}, I^{gt}_{t_k}) + \lambda_1 \mathcal{L}_{flow} + \lambda_2 \mathcal{L}_{sds} + \lambda_3 \mathcal{L}_{other}$

**end**

---

A detailed pseudo code for our flow supervision can be found in Algorithm 1. We extract the projected Gaussian dynamics and obtain the final Gaussian flow by rendering these dynamics. Variables including the weights and top-$K$ indices of Gaussians per pixel (as mentioned in implementation details of our main paper) are calculated in CUDA by modifying the original CUDA kernel codes of 3D Gaussian Splatting (Kerbl et al., 2023). And Gaussian flow $flow^G$ is calculated by Eq.8 with PyTorch.

In our 4D generation experiment, we run 500 iterations of static optimization to initialize 3D Gaussian fields with a batch size of 16. The Tmax in SDS is linearly decayed from 0.98 to 0.02. For dynamic representation, we run 600 iterations with batch size of 4 for both DG4D (Ren et al., 2023) and ours. The flow loss weight $\lambda_1$ in Eq. 11 of our main paper is 1.0.

Our method slightly decreases speed and increases memory only in training stage but not in the inference stage because our flow supervision is only for training a better/robust deformation field or other 4DGS designs and then will not be needed in inference stage. The training speed for DG4D is around 1.4it/s while it then becomes around 2.2it/s with our flow supervision. And the difference between training speeds with (around 2.5s/it) and without (around 2.2s/it) our flow supervision for RT-4DGS is marginal. Even with more memory footprint by tracking per-pixel gradients for Gaussians, a single 30GB GPU is adequate for reproducing all our results. In our 4D novel view synthesis experiment, we follow RT-4DGS(Yang et al., 2023c) except that we add our proposed flow supervision for all cameras. The flow loss weight $\lambda_1$ in Eq. 11 of our main paper is 0.5.

## A.2 More Visualization and Comparison in 4D Generation.

More comparisons between Gaussian flow $flow^G$ and optical flow $flow^o$ on rendered images are shown in Fig. 9. The first row of each example is the rgb frames rendered from a optimized 4D Gaussian field. We rotate our cameras for each time step so that the object can move as optimized and the camera is moving at the same time to show the scene from different angles. The second row of each example shows the visualized Gaussian flows. These Gaussian flows are calculated by the rendered images of consecutive time steps at each camera view, therefore, containing no camera motion in the flow values. The third row shows the estimated optical flows between the rendered images of consecutive time steps in each camera view. We use off-the-shelf AutoFlow (Sun et al., 2021) for the estimation. We can see that, enhanced by the flow supervision from the single-input view, our 4D generation pipeline can model fast motion such as the explosive motion of the gun hammer (see the last example in Fig. 9).

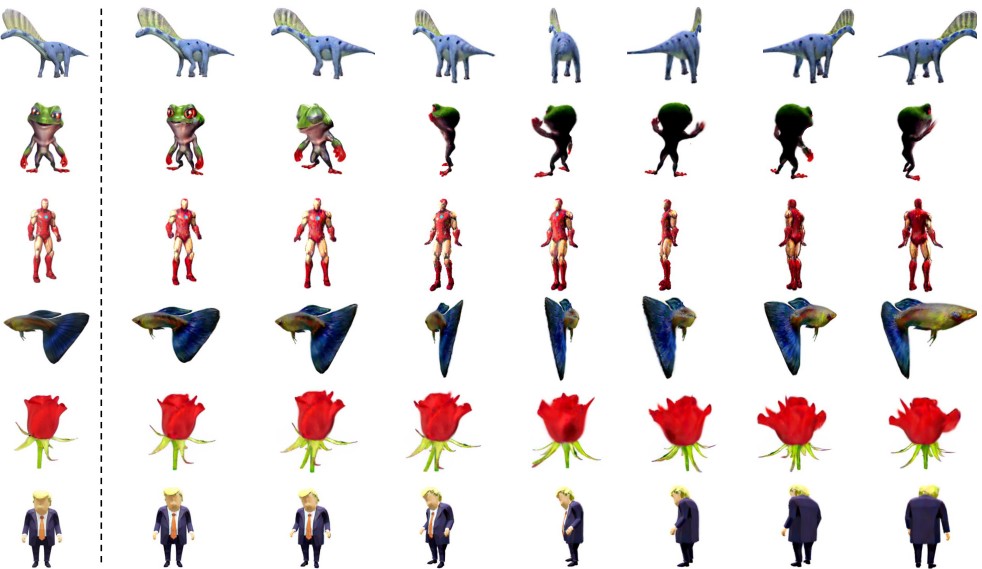

Figure 8: Qualitative results on Consistent4D dataset.

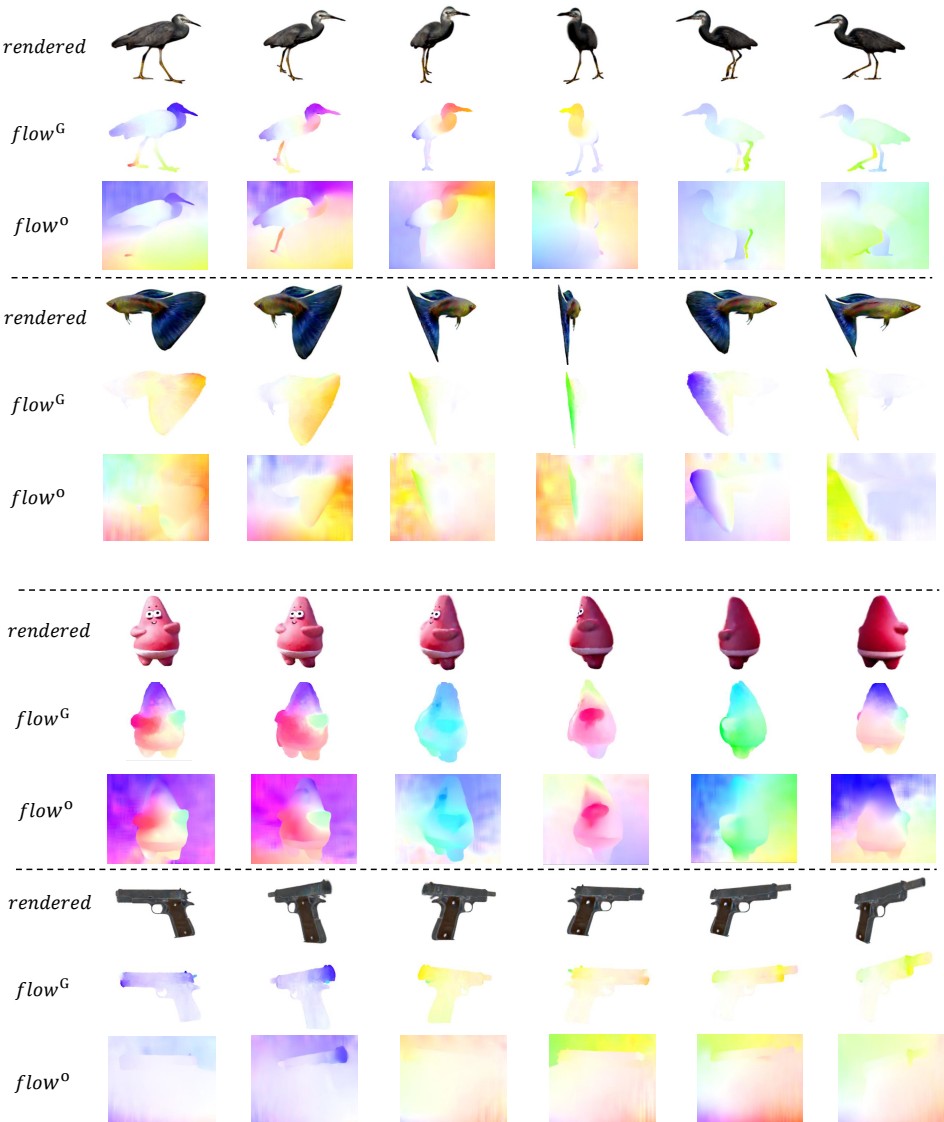

Figure 9: Visualization of Gaussian flow $flow^G$ and optical flow $flow^o$ on rendered sequences from different views.

### A.3 More Qualitative Results on the DyNeRF Dataset.

More qualitative results on DyNeRF dataset Li et al. (2022) can be found in Fig. 11, Fig. 12, Fig. 13 and our video.

### A.4 More Quantitative Ablations on the DyNeRF Dataset.

More quantitative ablations on the DyNeRF dataset Li et al. (2022) can be found in Tab. 6. If $\lambda_1$ is too small (e.g., $10^3$), the performance closely matches that of the baseline without flow supervision. Conversely, if $\lambda_1$. is too large (e.g., $10^3$), it leads to a performance drop in quantitative metrics such as PSNR. However, our flow supervision remains robust and effective when $\lambda_1$ is set within a moderate range. Another tunable parameter is the number of Gaussian K tracked per-pixel, we observed that increasing its value does not lead to significant performance gains, but it does slow down training due to the need to track a larger number

of Gaussians. Therefore, as mentioned in our Implementation Details, we use K = 20 to balance speed and effectiveness.

Table 6: Quantitative ablations on DyNeRF dataset Li et al. (2022). We report PSNR of our method with different number of Gaussians K tracked per-pixel and the flow loss weight $\lambda_1$.

|  | PSNR ↑ |
|---|---|
| Ours (K=0, $\lambda_1$=0.5) | 28.00 |
| Ours (K=10, $\lambda_1$=0.5) | 28.54 |
| Ours (K=40, $\lambda_1$=0.5) | **29.12** |
| Ours (K=20, $\lambda_1$=1e$^{-3}$) | 28.23 |
| Ours (K=20, $\lambda_1$=1e$^{3}$) | 27.35 |
| Ours (default:K=20, $\lambda_1$=0.5) | 28.99 |

### A.5 Flow Visualization in Dynamic Gaussian Fields

Note that dynamic 3D Gaussian (Luiten et al., 2023) provided a way to visualize 3D scene motions between consecutive frames, however, by tracking one "most influential" 3D Gaussian per pixel. This is neither efficient nor effective to be used in flow supervision, because the "most influential" 3D Gaussian for each pixel is determined by searching the nearest 3D Gaussian's center from tens of thousands of 3D Gaussian candidates with a virtual 3D point along pixel ray lifted with corresponding rendered depth. Also, the "most influential" Gaussian of a pixel might not even cover the same pixel but still be considered just because this Gaussian's center is the nearest one to the virtual point among all 3D Gaussians. We have also applied flow supervision in this way, but we find it has no observable benefit for rendering quality while resulting in slower training speed due to the per-pixel nearest search. On the other hand, RT-4DGS showed "render flow" in their paper only for visualization purpose and the detail were not clarified and the function was not enabled, please refer to their code, issue 1 and issue 2.

### A.6 Quantitative Results on the NeRF-DS Dataset.

When considering the cases with both camera motions and object motions, we have the relationship $warp(flow^o) = flow^G$, where $warp$ is the view warping of optical flow cased by camera motion and $flow^G$ is still the foreground object Gaussian dynamics. Therefore, we supervise the Gaussian flow with the warped optical flow by following camera and scene motion warping as (Li et al., 2021). We present qualitative comparisons of Deformable-3DGS (Yang et al., 2023d) with and without our flow supervision in Fig. 10.

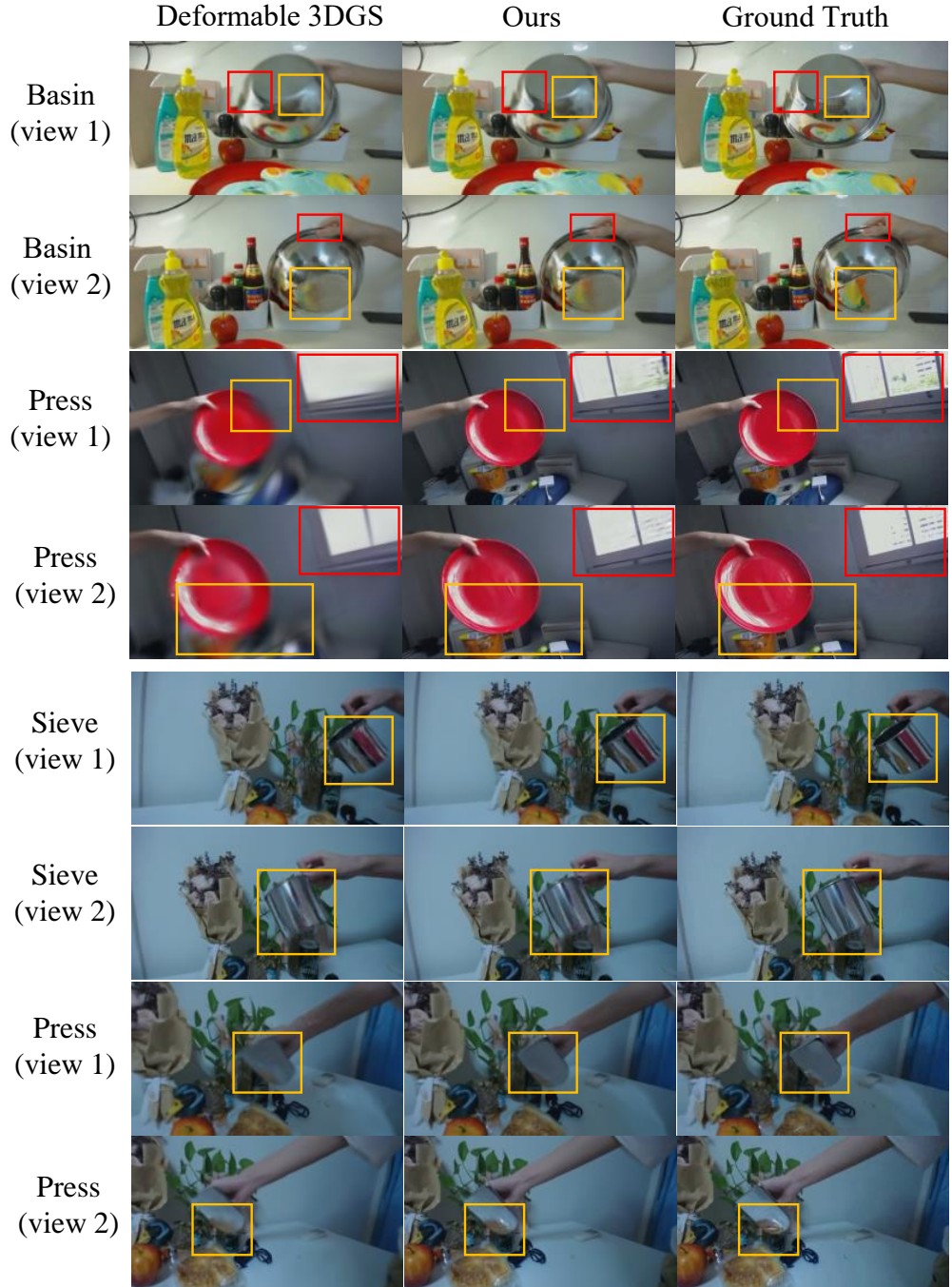

Figure 10: Qualitative comparisons on NeRF-DS dataset (Yan et al., 2023).

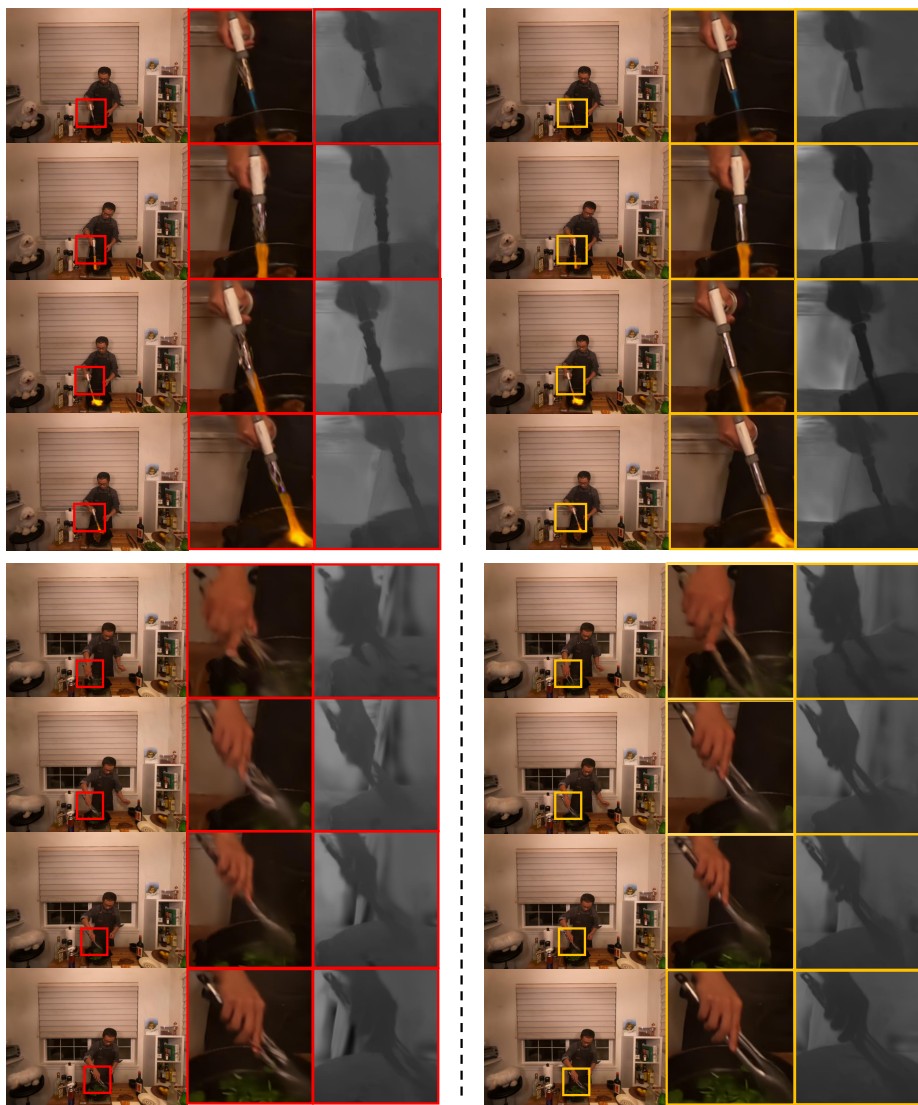

Figure 11: Qualitative comparisons on DyNeRF dataset (Li et al., 2022). The left column shows the novel view rendered images and depth maps of a 4D Gaussian method Yang et al. (2023c), which suffers from artifacts in the dynamic regions. The right column shows the results of the same method while optimized with our flow supervision during training.

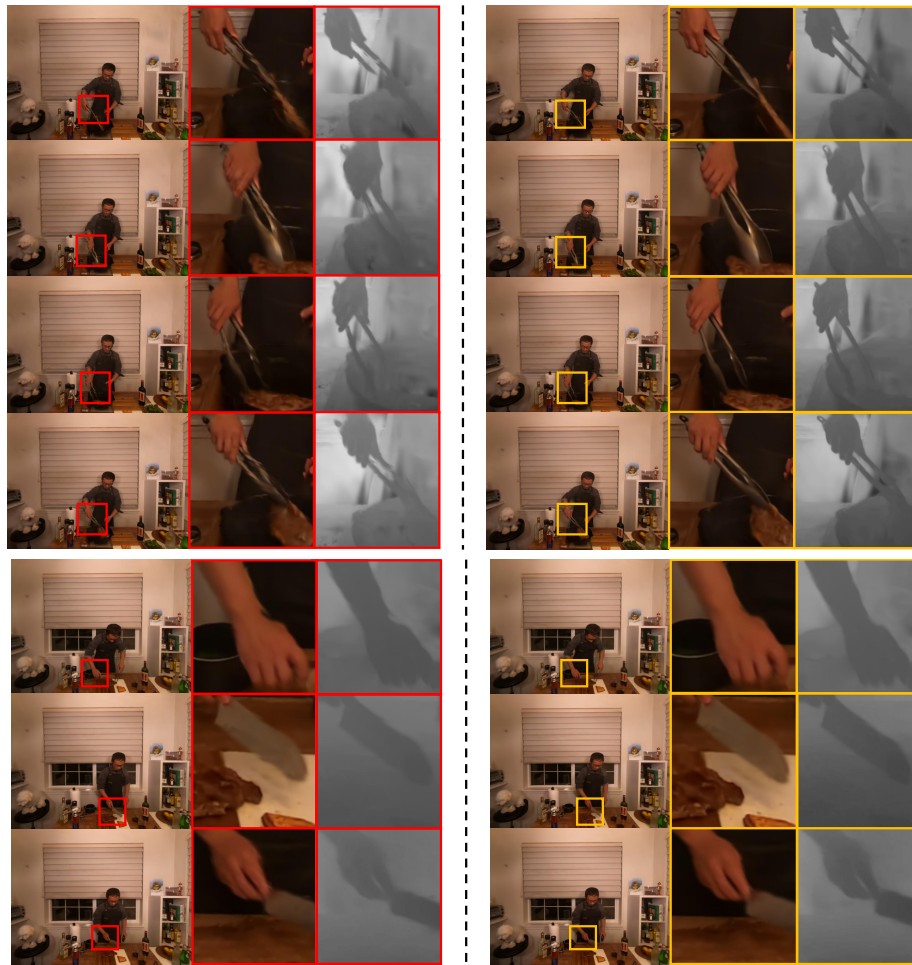

Figure 12: Qualitative comparisons on DyNeRF dataset Li et al. (2022). The left column shows the novel view rendered images and depth maps of a 4D Gaussian method (Yang et al., 2023c). While The right column shows the results of the same method while optimized with our flow supervision during training.

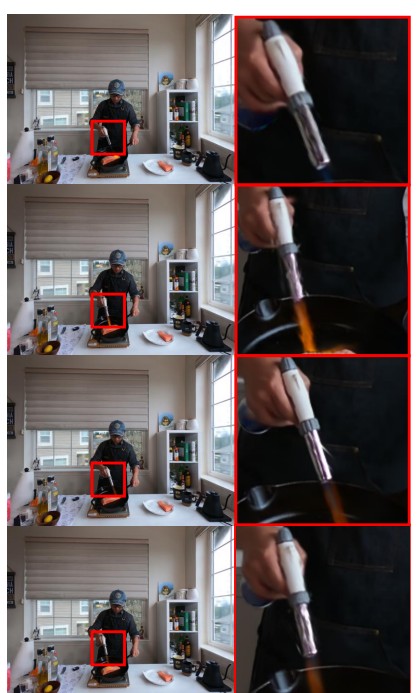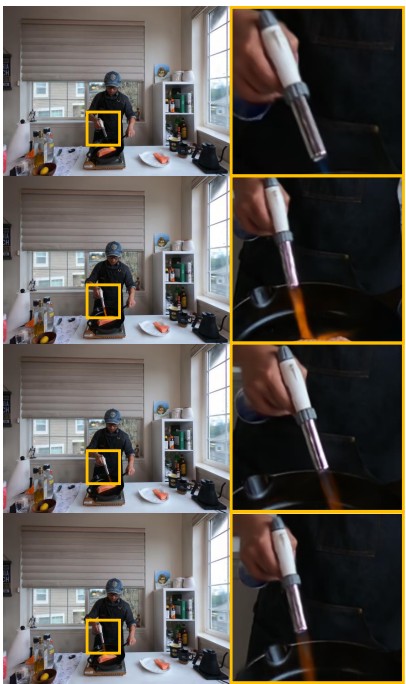

Figure 13: Qualitative comparisons on DyNeRF dataset (Li et al., 2022). Since the details of depth maps on *Flame Salmon* are hard to be recognized, we only compare the rendered images. The left column shows the novel view rendered images of a 4D Gaussian method (Yang et al., 2023c). While The right column shows the results of the same method while optimized with our flow supervision during training.

