# OpenReview forum: "GaussianFlow: Splatting Gaussian Dynamics for 4D Content Creation"
_TMLR — Accepted by TMLR_

### Review · Reviewer_gzjY · 2025-04-14

**Summary Of Contributions:**

This paper presents a method to calculate the flow of 4D Gaussian representations. Specifically, the flow of each pixel between two frames is calculated by weighting all the Gaussian movements on that pixel through alpha blending. The flow calculation acts as a plug-in module of flow supervision for 4D generation and reconstruction methods using Gaussian splatting. The authors demonstrate the effectiveness of their method by evaluating on different tasks, including 4D content generation and novel view synthesis.

**Audience:**

Yes

**Claims And Evidence:**

Yes

**Requested Changes:**

Please address the mentioned weaknesses in last section.

**Strengths And Weaknesses:**

Strengths:
- This paper proposed a generalizable and differentiable flow calculation method for 4D Gaussian splatting, which is valuable to the community.
- The authors presented a good evaluation on three datasets and two tasks, showcasing that their method can be applied to different 4D tasks, and could also deal with camera motion and object motion. The evaluation results also look great.
- Good analysis of overhead caused by the algorithm, which shows that the proposed method has minimal cost but provides relatively high improvement.

Weaknesses:
- The presentation of the paper might be further improved. The left and right part of Figure 2 is not described properly in the main text or caption. Lack of math symbols on the left of Figure 2.
- Is this method also works for monocular 4D reconstructions? What might be the challenge of doing that?
- I am curious about how large motion this method can handle since optical flow might be inaccurate for long-term dynamics.
- Lack of quantitive results on ablation study.
- Reference of most recent papers: GFlow, 4D-Rotor Gaussian splatting.

---

> ### Author Response · Authors · 2025-04-25
> **rebuttal**
>
> We sincerely thank the valuable comments and questions from the reviewer.\
> **The presentation of the paper might be further improved…**\
> Ans: Thanks for pointing out. We have provided more details in Section 3.2 of our revised version and clarify each notion in the caption of Fig.2. The intention of Fig. 2 (left) is to illustrate how the composited Gaussian primitives, projected from 3D space, contribute to the 2D optical flow from $t_1$ to $t_2$, corresponding to Eqs. (6)–(7) as described in our *Flow Composition* of Section 3.2. For clarity, we consider only two Gaussian primitives, indexed by i and j, between the two time steps $t_1$ and $t_2$. In contrast, Fig. 2 (right) shows how each individual Gaussian primitive morphs over time from$t_1$ to $t_2$, corresponding to Eqs. (3)–(5). \
> **Does this method also work for monocular 4D reconstructions...What might be the challenge of doing that** \
> Ans: Yes, monocular 4D reconstruction is indeed achievable with our method. To clarify terminology, we treat monocular 4D reconstruction as synonymous with 4D novel view synthesis (NVS) as shown in our paper, given that both are commonly evaluated using NVS metrics like PSNR due to the absence of 4D ground truth. The challenges of monocular 4D reconstruction typically include 1) occlusions, 2) motion ambiguities between static backgrounds and dynamic foregrounds, and 3) inaccurate camera pose estimation. The following discussion will detail how the use of optical flow addresses these specific challenges:
>
>   1)Occlusion and 2)motion ambiguities as demonstrated in one of the earliest works on dynamic NeRF [1], optical flow supervision serves as a crucial cue for resolving occlusions and motion ambiguity and enhancing both appearance and geometry quality, which is the idea also supported by the classical Lucas-Kanade method.
>       3) Regarding camera pose estimation, recent work such as FlowMap [2] has shown that camera parameters can be accurately estimated from off-the-shelf optical flow algorithms in a fully differentiable framework.
>
> These findings suggest that optical flow is a low-cost yet powerful motion cue for 4D reconstruction. Our method can serve as a bridge between the 4DGS representation and the aforementioned advances, contributing to more accurate and robust 4D reconstruction. Examples of applying optical flow supervision to 4DGS that are similar to ours can be found in [3,4].
>
> [1] Li et al. Neural scene flow fields for space-time view synthesis of dynamic scenes. CVPR 2021. \
> [2] Smith et al. FlowMap: High-Quality Camera Poses, Intrinsics, and Depth via Gradient Descent. 3DV, 2024. \
> [3] Wimmer et al. Gaussians-to-Life: Text-Driven Animation of 3D Gaussian Splatting Scenes. 3DV, 2024. \
> [4] Xie et al. Gaussian Splatting Lucas-Kanade. Arxiv, 2024. \
> **I am curious about how large motion this method can handle since optical flow might be inaccurate for long-term dynamics**\
> Ans:  Exactly. Our experiments with common 4D datasets haven't revealed significant failure modes since videos in these datasets only have small-to-moderate camera and object motions relative to the frame rate. But optical flow algorithms indeed struggle with resolving 3D ambiguities, including self-rotations, and are also vulnerable to occlusions and motion blur, particularly when inter-frame pixel correspondence is lost. Therefore, inaccurate optical flow estimation for long-term dynamics may deteriorate the performance of our method. \
> **Lack of quantitative results on ablation study.** \
> Ans: Thanks for pointing it out. We've updated the quantitative ablations in our revised version. Please refer to our appendix A.4 More Quantitative Ablations on the DyNeRF Dataset for more details. \
> **Missing references**\
> Ans: We thank the reviewer for these two excellent references. They are highly relevant to our work, and we have included them in the revised version.

---

### Review · Reviewer_bpt7 · 2025-04-16

**Summary Of Contributions:**

This paper introduces GaussianFlow, an innovative approach to connect the motion dynamics of 3D Gaussian representations to 2D pixel velocities, enabling effective supervision from optical flow for Gaussian-based dynamic scene reconstruction. The approach involves splatting 3D Gaussian motions onto the image plane, thus establishing a differentiable relationship between Gaussian dynamics and pixel optical flow. Experiments on various datasets demonstrate that GaussianFlow significantly enhances the quality of 4D novel view synthesis and 4D content generation, particularly in dynamic and challenging motion scenarios. The method also addresses the color drifting artifacts common in existing techniques, providing state-of-the-art results compared to contemporary 4D reconstruction methods.

**Audience:**

Yes

**Claims And Evidence:**

Yes

**Requested Changes:**

- Could the authors clarify how GaussianFlow’s performance degrades as the accuracy of the optical flow deteriorates? Are there mechanisms proposed or envisioned to mitigate such dependencies?
- Are there plans or potential directions for extending GaussianFlow to handle long-term temporal consistency across multiple frames rather than just consecutive frames?
- How significantly does the choice of initialization methods (e.g., DreamGaussian or One-2-3-45) influence the GaussianFlow optimization process and the final visual results? Could this approach integrate more adaptive or data-driven initialization techniques?

**Strengths And Weaknesses:**

Strengths:
+ The introduction of GaussianFlow, bridging Gaussian splatting dynamics directly with optical flow supervision, represents a meaningful technical advancement. The approach addresses the challenge of directly supervising 3D Gaussian motions from 2D pixel data.
+ Experimental results show improvements, especially in highly dynamic scenarios where previous methods typically suffer from motion ambiguities and visual artifacts. This indicates robustness and practical applicability.
+ The method leverages efficient CUDA-based implementations, ensuring minimal computational overhead despite its sophisticated supervision mechanism. This is critical for practical deployment and scalability in real-time applications.

Weaknesses:
- The method seems to heavily rely on the accuracy of optical flow estimation. Inaccuracies or failures in optical flow estimation could directly degrade GaussianFlow’s performance, especially in scenes with repetitive textures or fast motion.
- The current approach primarily supervises short-term optical flows between consecutive frames. This may limit effectiveness in long-duration sequences, potentially causing cumulative errors or drift over extended temporal windows.
- While the evaluation includes high-quality synthetic datasets, there is limited demonstration of the method's robustness in uncontrolled, real-world scenarios, which may exhibit more severe challenges such as lighting changes, occlusions, and noise.

---

> ### Author Response · Authors · 2025-04-25
> **rebuttal**
>
> We sincerely thank the valuable comments and questions from the reviewer.\
> **heavily rely on the accuracy of optical flow estimation** \
> Ans: The effectiveness of GaussianFlow does indeed depend on the accuracy of optical flow estimation. We acknowledge that optical flow algorithms can fail in certain extreme cases such as very fast motions and self-rotations, which remain an open challenge in the computer vision community and beyond the scope of our paper.
> However, off-the-shelf optical flow methods such as RAFT and VideoFlow are generally robust, even in challenging scenarios involving moderate to fast motion. We refer to the training datasets and demo results of RAFT and VideoFlow for their capabilities. Notably, the use of optical flow supervision for dynamic scene modeling has been explored since Dynamic NeRF [1] as a means to reduce motion ambiguity. The primary focus of our work is not on selecting the optimal optical flow algorithm, but on proposing a novel formulation to quantify the 3D–2D correspondence of Gaussian dynamics. This is a non-trivial task due to the inherent properties of 3DGS, including non-isotropic local radiance parameterized by scaling and rotation factors, as well as the use of an approximate 3D-to-2D projection rather than the true perspective projection as in NeRF.  \
> [1] Li et al. Neural scene flow fields for space-time view synthesis of dynamic scenes. CVPR 2021. \
> **primarily supervises short-term optical flows between consecutive frames**\
> Ans:  Yes, as noted in our *Future Work* section, exploring the benefits of long-term flow supervision is left for future investigation. We follow the common way [1] of applying optical flow calculated from consecutive frames to supervise dynamic scene representation without extensive hyperparameter tuning or reliance on the most advanced off-the-shelf algorithms.\
> **there is limited demonstration of the method's robustness in...**\
> Ans: In addition to synthetic videos as Consistent4D Dataset, we demonstrate the effectiveness of our method on real-world videos from the Plenoptic Video Dataset and the NeRF-DS Dataset. These widely used benchmarks include challenges such as occlusions, noise, and specular effects, making them suitable for evaluating methods in dynamic scene reconstruction and novel view synthesis.\
> **how GaussianFlow’s performance degrades as the accuracy of the optical flow deteriorate**\
> Ans: Our method incorporates two mechanisms to mitigate the impact of inaccurate optical flow estimation. 1) First, it is designed to ignore noisy optical flow in empty regions. As illustrated in Fig. 7, white or empty areas across consecutive frames often exhibit unreliable optical flow $flow^o$, which does not correspond to either camera or object motion. Since our method establishes 3D–2D correspondences between 3D Gaussian primitives and their 2D projections, regions of empty space without any projected Gaussians are naturally unaffected by noisy optical flow supervision. As a result, our predicted flow $flow^G$ has no Gaussian flow estimated in background regions. 2) Second, we have a pre-set threshold to filter out noisy optical flow in implementation. This pre-set threshold is the mean value of the entire optical flow map at the current timestep. The optical flow below this value will be discarded and only photometric loss is considered in these areas.\
> **extending GaussianFlow to handle long-term temporal consistency**\
> Ans: As noted in our *Future Work* section, exploring the benefits of long-term flow supervision is left for future investigation. However, our formulation remains consistent regardless of whether short-term or long-term flow supervision is applied and the only change is on the optical flow estimation side. For instance, optical flow at a given time step can be warped from future frames using a sliding window, and Gaussian motions between any two timesteps (not limited to consecutive ones) can be derived from the deformation field without requiring any additional technical modifications. In practice, the choice between short-term and long-term flow estimation often depends on optical flow algorithms themselves. VideoFlow, for instance, offers two pre-trained checkpoints: one tailored for short input sequences and another designed to capture long-term motion, which could lead to improved performance. \
> **initialization methods**\
> Ans:  Widely adopted 3DGS initializations are SfM points. However, the monocular videos in the Consistent4D dataset lack SfM points as there is no camera motion. Therefore, as shown in Fig. 3, we initialize 3DGS on the first input frame with photometric loss and Zero-1-2-3 supervision, also known as SDS supervision, which is a popular data-driven initialization employed in DreamGaussian and One-2-3-45 for 3DGS or mesh reconstruction. Since this 3DGS initialization strategy is neither proposed nor emphasized in our work, we didn't present it as one of our contributions.

---

### Review · Reviewer_tqNG · 2025-04-26

**Summary Of Contributions:**

This paper introduced a novel method considering the flow of the Gaussian ellipsoids. The whole flow algorithm can mainly be divided into two parts: the flows of a single Gaussian ellipsoid (through normalization and unnormalization) and multiple Gaussian ellipsoids (through flow composition by alpha-blending). The idea is simple but effective via the shown experiments. Also, they test their proposed method on two tasks: 4D generation and 4D NVS.

**Audience:**

Yes

**Broader Impact Concerns:**

Nothing.

**Claims And Evidence:**

Yes

**Requested Changes:**

1. I understand the key contribution of this paper is not simply flow supervision, but I highly suggest that the author should compare it with some more flow supervision methods to further demonstrate the effectiveness of their method. For example, 'GFlow: Recovering 4D Worlds from Monocular Video' and 'Motion-aware 3D Gaussian Splatting for Efficient Dynamic Scene Reconstruction'.

**Strengths And Weaknesses:**

**Strengths**
1. Such a plug-in flow supervision module is effective and can be easily integrated into different works.
2. The authors demonstrate the effectiveness of their proposed method on 4D generation and 4D NVS tasks.
3. Extensive experiments show their superiority.

**Questions**
1. Can the proposed method handle the scenes with specular surface objects?

---

> ### Author Response · Authors · 2025-04-26
> **rebuttal**
>
> We sincerely thank all the valuable comments and insightful questions from the reviewer. \
> **Can the proposed method handle the scenes with specular surface objects** \
> Ans: Yes, please refer to Fig. 4, Figs. 11–13, and the supplementary video for qualitative results, and to Tab. 1 for quantitative results on the DyNeRF dataset. Notably, many moving objects, such as the torch gun, clip, and pan are metallic and exhibit view-dependent specular highlights. And our method can still improve the rendering quality and geometry reconstruction. Other objects in the NeRF-DS dataset also show specular effects, please refer to Tab. 2 and Fig. 10 for the corresponding quantitative and qualitative results, respectively.
>
> **compare it with some more flow supervision methods** \
> Ans: We thank the reviewer for these two excellent references and we have included them in the revised version. Specifically, GFlow involves multiple motion prior such as point tracking, optical flow and depth supervision, leading to superior performance as shown in the paper. The key difference in using optical flow supervision between GaussianFlow and GFlow is as follows: GaussianFlow directly establishes correspondences between Gaussian dynamics and per-pixel optical flow, allowing the gradient of optical flow with respect to Gaussian parameters to be computed in a closed form. This is similar to [1], which aggregates sample points along a ray from the previous timestep, moves them to new positions at the current timestep driven by a deformation field or scene flow, and then projects the aggregated point into 2D to compute the offset relative to the original pixel location. The key differences between [1] and our approach are: (1) they composite the motions of 3D sample points before projecting them into 2D, whereas we composite the motions of 2D Gaussians directly to calculate 2D flow; and (2) we explicitly account for the shapes of Gaussian primitives, whereas NeRF sample points do not have intrinsic shapes or local radiance properties as 3D Gaussians do. Therefore, we observed quality improvements in 4D novel view synthesis similar to those  in [1]. For more details, please refer to their [implementation](https://github.com/zhengqili/Neural-Scene-Flow-Fields/blob/main/nsff_exp/run_nerf_helpers.py#L567C5-L567C25) for the details.
>
> In contrast, GFlow applies optical flow supervision in the same manner as photometric supervision. Although no official code for GFlow is publicly available, we implemented its formulation and applied it to the DyNeRF dataset. We observed marginal improvement (less than 0.1dB in terms of PSNR) over the baseline method RT-4DGS. A potential reason is that the gradient of optical flow with respect to any specific Gaussian motion is ambiguous: pixel-wise optical flow often corresponds to many Gaussian motions between two timesteps, and the processes of Gaussians “moving into” or “moving out of” a pixel further introduce ambiguity in flow correspondence. Therefore, the benefit of optical flow supervision in GFlow partially overlaps with that of photometric supervision.
>
> For Motion-aware 3D Gaussian Splatting for Efficient Dynamic Scene Reconstruction (M3GS), both M3GS and our work report quantitative results on the DyNeRF dataset. According to Table 2 of the M3GS paper, M3GS-I achieves an average PSNR of **33.27** dB, and M3GS-D achieves **32.09** dB across the cook_spinach, cut_roasted_beef, and sear_steak sequences. In comparison, our method achieves an average PSNR of **33.97** dB (please refer to our Tab.1), outperforming M3GS.
>
> [1] Li et al. Neural scene flow fields for space-time view synthesis of dynamic scenes. CVPR 2021.

---

> ### Comment · Reviewer_tqNG · 2025-04-28
> **Response**
>
> Thanks for the authors' answers. It addresses some of my concerns. However, I think the DyNeRF dataset contains only a few specular objects, or we can say the specular objects in the scenes are not that large and obvious, so somehow I think it cannot concretely support the authors' statement about the model's performance on the specular objects. I think the NeRF-DS[1] dataset is famous for its large and specular objects in the scenes without too many scenarios. So I suggest the authors conduct experiments on the NeRF-DS dataset to further demonstrate the superior performance of their proposed model.
>
>
>
> [1] Yan, Zhiwen, Chen Li, and Gim Hee Lee. "Nerf-ds: Neural radiance fields for dynamic specular objects." In Proceedings of the IEEE/CVF Conference on Computer Vision and Pattern Recognition, pp. 8285-8295. 2023.

---

> > ### Author Response · Authors · 2025-04-28
> > **NeRF-DS dataset**
> >
> > We thank the reviewer's response.
> > Please refer to Tab. 2 and Fig. 10 for the corresponding quantitative and qualitative results on NeRF-DS dataset, respectively.

---

### Decision · Action_Editor_JsPg · 2025-06-09

**Recommendation:** Accept as is

**Audience:**

Yes

**Audience Explanation:**

The overall area is of strong interest to the community, and the method is likely to be of interest to people in the community.

**Claims And Evidence:**

Yes

**Claims Explanation:**

After the response, all three reviewers believe that the submissions' claims are supported by accurate, convincing and clear evidence. The revised paper is clearly ready for acceptance.